# A mathematical model for persistent post-CSD vasoconstriction

**Shixin Xu**[1,2,3], **Joshua C. Chang** [4,5,6]*, **Carson C. Chow** [4], **KC Brennan**[7], **Huaxiong Huang**[2,3,8]*

**1** Duke Kunshan University, 8 Duke Ave., Suzhou, China, **2** Department of Mathematics and Statistics, York University, Toronto, Ontario, Canada, **3** Centre for Quantitative Analysis and Modeling (CQAM), The Fields Institute for Research in Mathematical Sciences, 222 College Street, Toronto, Ontario, Canada, **4** Laboratory of Biological Modeling, NIDDK, National Institutes of Health, Bethesda Maryland, United States of America, **5** Epidemiology and Biostatistics Section, Rehabilitation Medicine Department, The National Institutes of Health, Bethesda Maryland, United States of America, **6** mederrata, Columbus Ohio, United States of America, **7** Department of Neurology, University of Utah, Salt Lake City, Utah, United States of America, **8** Research Center for Mathematics, Advanced Institute of Natural Sciences, Beijing Normal University (Zhuhai), Guangdong, China

* josh.chang@nih.gov (JCC); hhuang@yorku.ca (HH)

**Data Availability Statement:** Source code is available at https://github.com/xsxztr/xsxztr-Post-CSD-code-with-Joshua-C.-Chang.

**Funding:** This work was supported in part by the Intramural Research Programs of the National

## Abstract

Cortical spreading depression (CSD) is the propagation of a relatively slow wave in cortical brain tissue that is linked to a number of pathological conditions such as stroke and migraine. Most of the existing literature investigates the dynamics of short term phenomena such as the depolarization and repolarization of membrane potentials or large ion shifts. Here, we focus on the clinically-relevant hour-long state of neurovascular malfunction in the wake of CSDs. This dysfunctional state involves widespread vasoconstriction and a general disruption of neurovascular coupling. We demonstrate, using a mathematical model, that dissolution of calcium that has aggregated within the mitochondria of vascular smooth muscle cells can drive an hour-long disruption. We model the rate of calcium clearance as well as the dynamical implications on overall blood flow. Based on reaction stoichiometry, we quantify a possible impact of calcium phosphate dissolution on the maintenance of $F_0F_1$-ATP synthase activity.

## Author summary

This manuscript links calcium phosphate cluster dissolution in mitochondria to oscillations in vascular smooth muscle cells in putting forward an explanation for deranged vascular dynamics following cortical spreading depression (CSD). CSD is an extraordinary phenomenon in the brain triggered in many adverse events such as migraine with aura, stroke, and traumatic brain injury. As a propagating wave phenomenon, researchers have concentrated on CSD's acute spreading phase. However, CSD exhibits stereotyped post-acute dynamics, and is a candidate for explaining longer-lasting derangement in brain activity.

Our explanation for the post-acute dynamics relates to superphysiological calcium buffering in vascular smooth muscle cells. In CSD, extracellular calcium floods into cells

Institutes of Health Clinical Center and the National Institute of Diabetes and Digestive and Kidney Diseases, NIH R01 NS 102978 and NS 104742, NSERC, the Fields Institute, DKU startup, and FSTA fund. The funders had no role in study design, data collection and analysis, decision to publish, or preparation of the manuscript.

in the brain, including neurons, vascular cells, and glial cells. When cells are at extremely high calcium loads, mitochondria buffer excess calcium by forming calcium phosphate precipitate species. The existence of these species stabilizes the free ionic calcium concentration within mitochondria to a set point determined by thermodynamic equilibrium of the solvation process. This concentration is elevated relative to normal mitochondria concentration—we explore the implications of the stable elevation on calcium dynamics within vascular smooth muscle cells and relate these dynamics to macroscopic physiological behavior of the vasculature after CSD.

## Introduction

Cortical spreading depression (CSD) [1, 2, 3, 4, 5], is a slow-moving propagating wave in cortical gray matter tissue. Its presence is widespread in distressed states including traumatic brain injury, stroke, and migraine with aura [6, 7, 8, 9]. CSD presents at several timescales. The two-minute timescale of changes occurring as the wave passes is fairly well-studied, as is the transient quiescence of neuronal activity in its wake [10]. However, an hour-long period characterized by persistent vasoconstriction [11] and derangement in neurovascular coupling [12] (NVC) remains relatively enigmatic. Neurovascular coupling refers to the dynamical matching of cerebral blood flow to changes in metabolic need induced by neuronal activity [13, 14]. This longer timescale is clinically relevant, for instance in concussion syndrome after traumatic brain injury [15].

Here, we advance a mathematical model for the persistent vasoconstriction based on the following experimental observations: 1) During the acute phase of CSD, extracellular ionic calcium drops by 90% while intracellular calcium elevates to super-physiological concentrations [16]. 2) Vasoconstriction is ultimately controlled by calcium dynamics in vascular smooth muscle cells [14]. 3) Blockage of mitochondrial permeability transition pore (mPTP) formation in the acute phase of CSD prevents the persistent vascular phenotype [17]. The mPTP may be allowing large amounts of calcium to enter mitochondria. 4) Mitochondria are limitless reservoirs for calcium absorption through precipitation of calcium phosphate species [18] —in vascular smooth muscle precipitates may act as a source for calcium. 5) Supersatured solutions balance aggregation and dissolution to settle at a lower thermodynamically stable concentration [19, 20] that is lower than the total concentration, yet still elevated. 6) The rates of exchange mechanisms depend on the active soluble concentration of calcium rather than the total concentration.

Overall, we consider the implications of the formation of clusters—with their implied steady state elevation of mitochondria calcium—on cytosolic calcium dynamics within vascular smooth muscle cells. In this model, the saturation concentration rather than the total concentration of calcium in the matrix determines the exchange rate between mitochondria and the cytosol. Using this model, we demonstrate that it can account for the observed hour-long timescales while mimicking the macroscopic phenotype of the pial artery constriction of Figure 4A of [12].

## CSD: Setting the stage

Mechanistically, CSD is a self-propagating metabolic wave that involves all electrically polar cells in the affected tissue—neurons, glia, and vascular cells. Although synaptic vesicle release occurs during CSD, the primary driver of CSD seems to be extra-synaptic activity through what has been termed "volume-transmission," where CSD is best-described as a reaction-

diffusion phenomenon [2]. In CSD, widespread cellular depolarization occurs in the presence of focally large extracellular concentrations of potassium and excitatory neurotransmitters such as glutamate at the wavefront [21, 22]. In turn, this depolarization induces the release of more potassium and glutamate, sustaining wave propagation. The vasculature is an active participant. There is evidence that vascular cells conduct the wave along their major axes [23, 24]. Besides conducting the wave, arteries in the brain respond to the elevated potassium by constricting [25], although the change in oxygen supply does not itself appear to have much effect on the properties of the propagating wavefront [26].

The recovery of the tissue following a CSD is biphasic. The first phase corresponds to the recovery at the acute wavefront and lasts approximately two minutes from an initial elevation of extracellular potassium and glutamate to a general quiescence of activity. At the end of this phase, extracellular potassium, sodium, and glutamate recover to near their pre-CSD levels. The second phase involves the recovery from tissue-scale derangements. Several minutes after the end of the acute phase, cerebral tissue undergoes changes [27]. The cerebral vasculature exhibits dysregulation in NVC [12, 28]. Beyond the mismatch of blood flow to neuronal metabolic need, overall blood flow is reduced [29] as cerebral vasculature notably constricts relative to baseline. The recovery from this state occurs on the timescale of an hour [12].

In the acute phase, extracellular calcium concentration drops from a baseline of over 1 mM [30] to under 0.1 mM [30]. The actual net movement of calcium is even more drastic when accounting for the fact that the extracellular space shrinks in half due to the increase in cellular volume from the increased osmotic pressure [31]. These two factors combined imply that approximately 95% of the initial extracellular calcium transfers into cells during the acute phase.

Lacking vascular smooth muscle calcium measurements of calcium dynamics during CSD, we rely on neuronal measurements, where cytosolic calcium elevation has been observed [21]. In experiments in the absence of external calcium, there is no elevation of neuronal intracellular calcium [32], suggesting an extracellular source of calcium. We know from the fact that the vasculature constricts in the acute phase that cytosolic calcium is elevated throughout its duration. Some of this calcium is presumably buffered within mitochondria. Under normal circumstances, the amount of calcium entering the matrix is limited by the uptake of the mitochondrial calcium uniporter (MCU) [33]. However, in CSD, mitochondrial depolarization [34] induces the formation of the mitochondrial permeability transition pore (mPTP), a non-specific pore across the mitochondrial membrane. Blockage of mPTP formation has been shown to prevent the second phase [17]. In typical settings, formation of this pore implies the exit of calcium out from an overloaded mitochondria into the cytosol [35]. CSD is not a typical situation. In CSD, this pore forms while cytosolic calcium is elevated within the cytosol. Hence, the mPTP provides a mechanism for large-scale calcium entry into the matrix.

As opposed to the acute stage [36], vasoconstriction in the second phase is not due to extracellular potassium [12], which would suggest an extracellular voltage-dependent calcium source for inducing vasoconstriction. In this manuscript, we propose a vasogenic source of vasoconstrictive free calcium. Through mathematical modeling, we examine the hypothesis that slow calcium release from mitochondrial calcium stores of vascular origin can account for the behavior seen in this longer phase of recovery.

## Mitochondrial calcium uptake

In CSD, cells in the gray matter absorb calcium. Vascular smooth muscles cells are known to be adept at absorbing calcium, with excess calcium mineralizing in the mitochondrial matrix [37]. Mitochondria are calcium buffers of last resort. The primary calcium uptake mechanisms

for mitochondria are the MCU and the reversible mitochondrial sodium-calcium exchanger (NCLX). In normal physiological states, mitochondria are likely not important buffers of calcium [33]. The endoplasmic (sarcoplasmic in muscle) reticulum (ER) offers a stronger but limited basin for calcium sequestration. However, mitochondria are the dominant mechanism for calcium sequestration in highly distressed cell states due to their virtually limitless capacity for calcium buffering [18]. The buffering capacity is not due to presence of binding proteins as in other cell compartments. Rather, this capacity arises from interactions between ionic calcium, inorganic phosphate, and protons that result in nucleation and growth of calcium orthophosphate clusters [20, 38]. Here, nucleation refers to the physical process by which ordered states like clusters arise from less-ordered states like supersaturated solutions, where supersaturation is a state where solutes are crowded in solution past a critical concentration where it is thermodynamically favorable to cluster.

A limited mechanism for inhibiting nucleation exists within mitochondria. Inorganic phosphate exists both in its free monomeric forms ($PO_4^{3-} \rightleftharpoons HPO_4^{2-} \rightleftharpoons H_2PO_4^- \rightleftharpoons H_3PO_4$) and in polymeric forms as inorganic polyphosphate (polyP [39]) molecules, held together by the same phosphoanhydride bonds that endow ATP with high chemical potential energy. PolyP is thought to inhibit the formation of clusters [40], by binding to precursors called prenucleation clusters [41, 42].

## Calcium phosphate nucleation and dissolution

In solutions, solvation describes the complicated interactions between a solvent and solute particles. The observed dynamics result as an interplay between two competing drives—solute-solvent interactions, and solute-solute interactions. A solution is said to be at saturation when these drives are balanced, and at supersaturation when solute-solute interactions dominate. When there is an excess of solute relative to the saturation point, precipitation is thermodynamically favored. However, precipitation is often limited by kinetic considerations. The rate-limiting step in this process is nucleation, which depends on the chemical properties of the solution.

The chemical interactions between calcium, inorganic phosphate, and other salts in aqueous solution are complex [20, 38, 43, 44]. When calcium and phosphate are at sufficiently high concentrations, they co-aggregate to form complexes of varying stoichiometry [45], first as prenucleation clusters [46], and then as stable nuclei. Prenucleation clusters offer a quicker path to particle condensation than predicted according to nucleation kinetics. For calcium phosphate in the process of condensation, protons are liberated into the solution decreasing the pH [20].

Biological solutions exist in a baseline state of supersaturated calcium and phosphate, maintaining this state actively through the action of nucleation inhibitors. These inhibitors prevent runaway calcification from occurring. The main inhibitor of calcium phosphate nucleation in extracellular bodily fluids is fetuin-A [47, 48, 49], which has strong affinity for prenucleation clusters, thereby exponentially increasing the concentration of calcium that is able to exist stably in solution [50]. In the presence of nuclei, a reduction of free calcium or phosphate concentration below the thermodynamic saturation point results in dissolution which replenishes the free concentration. Elevations result in aggregation which reduces the free concentration. Hence, the saturation point is a thermodynamically stable steady state.

We assume that the mitochondrial matrix exists at quasi-steady equilibrium at saturation and examine the implications of this persistent calcium elevation on cellular calcium dynamics. To set the stage for this state, we assume that large quantities of calcium have entered the mitochondria during the acute phase of CSD, through the mPTP. After closure of the mPTP and commencement of the aggregation process, we assume that the mitochondrion is able to

return to a stable baseline pH, where the usual exchange mechanisms act to expel the excess calcium stores.

## Mathematical model

The chief objective of our model is to understand the dynamics of free ionic calcium in the cytosol as calcium leaking from mitochondrial stores interact with the various cellular mechanisms for calcium homeostasis.

We denote concentrations using square brackets (e.g. [Ca$^2$+] corresponds to the concentration of free ionic calcium), generally parameterized in units of micromolar ($\mu$-moles per liter). The concentrations are specific to sub-cellular or extracellular compartments—we keep track of these using subscripts. The compartments modeled are the extracellular space (ecs), the cytosol (cyt), the mitochondrial matrix (mit), and the endo(sarco)plasmic reticulum (er). Concentration fluxes (of calcium) between the compartments are denoted by $J$ and assumed to be oriented so that positive $J$ is relative to positive changes in cytosolic calcium concentration. Additionally, all $J$ fluxes are scaled to the cytosolic volume.

The intracellular compartments have volumes denoted by $V_{cyt}$, $V_{mit}$, $V_{er}$. Note that these volumes are overall volumes within a single cell—the mitochondrial volume scales with the number of total mitochondria. From these volumes arise the volume ratios

$$r_{er}^{cyt} = \frac{V_{cyt}}{V_{er}} = 10, \quad r_{mit}^{cyt} = \frac{V_{cyt}}{V_{mit}} = 13.6, \tag{1}$$

where the cytosolic volume is assumed to be 0.7 pL and the mitochondrial volume ratio is taken from [51]. We assume that calcium is well-mixed within each compartment, denoting the compartmental concentrations as [Ca$^{2+}$]$_{cyt}$, [Ca$^{2+}$]$_{mit}$ and [Ca$^{2+}$]$_{er}$ accordingly. A schematic of our model is presented in Fig 1.

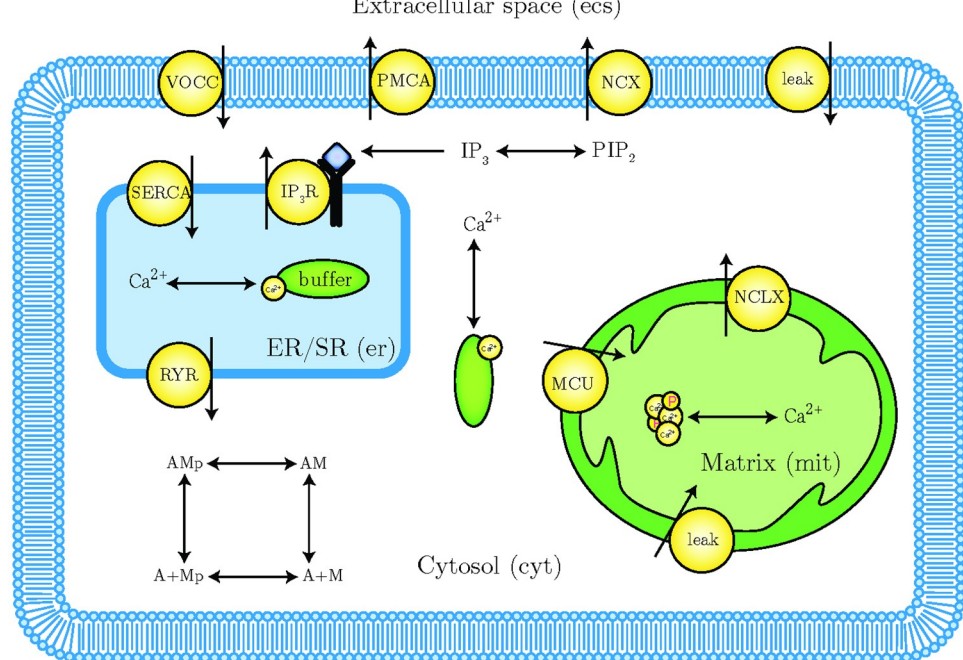

**Fig 1. Model schematic showing compartments, calcium exchanges, and other reactions specifically considered.** Note that only calcium ions are considered in our model so we ignore counter-transporters such as sodium. Most of the exchange mechanisms are bidirectional—direction that is relevant to the scope of our model is depicted.

## Cytosol

The concentration of calcium in the cytosol obeys mass balance through the ordinary differential equation

$$\frac{d[Ca^{2+}]_{cyt}}{dt} = \frac{1}{b_{cyt}}(J_{er \to cyt} + J_{mit \to cyt} + J_{ecs \to cyt} + J_{cyt \to cyt}). \tag{2}$$

The flux terms account for exchange between the cytosol and other compartments, as well as reactions within the cytosol. The term in the denominator of Eq 2 models calcium buffering and is given by

$$b_{cyt} = 1 + \frac{K_{CaM}[CaM]_{cyt}}{(K_{CaM} + [Ca^{2+}]_{cyt})^2} + \frac{K_B[B]_{cyt}}{(K_B + [Ca^{2+}]_{cyt})^2}. \tag{3}$$

It is due to the rapid buffer approximation (RBA) [52], a form of the total quasi-steady-state approximation [53], which accounts for the effects of binding proteins (parameters given in Table 1). Note that we explicitly distinguish between (active) free ionic calcium and (inactive) bound calcium in our model. Consistent with this approximation, the total (free + bound) concentration of calcium follows

$$[Ca^{2+}]_{cyt,tot} = [Ca^{2+}]_{cyt}\left(1 + \frac{K_{CaM}[CaM]_{cyt}}{K_{CaM} + [Ca^{2+}]_{cyt}} + \frac{K_B[B]_{cyt}}{K_B + [Ca^{2+}]_{cyt}}\right). \tag{4}$$

where all quantities are fixed and listed in Table 1.

**Table 1. Cytosol-specific model parameters.** Buffer parameters rescaled from [55] to yield a baseline buffering ratio of 1%, consistent with [56]. [Myo] made small to have little impact relative to other buffers.

| Parameter | Value | Description | Notes |
|---|---|---|---|
| Buffer | | | |
| $K_{CaM}$ | $0.260\,\mu M$ | Dissociation rate of calmodulin buffer | this work |
| $[CaM]_{cyt}$ | $30\,\mu M$ | Concentration of calmodulin site | |
| $K_B$ | $0.530\,\mu M$ | dissociation rate of "other" buffer | |
| $[B]_{cyt}$ | $30\,\mu M$ | Concentration of "other" buffers | |
| Myosin | | | |
| $[Myo]$ | $10\,\mu M$ | Total concentration of myosin | this work |
| $\gamma_{cross}$ | $17\,(\mu M)^{-3}\,s^{-1}$ | | [54] |
| $K_1$ | $\gamma_{cross}[Ca^{2+}]_{cyt}^3$ | Rate in changing from M to Mp | |
| $K_2$ | $0.5\,s^{-1}$ | Myosin dephosphorylation rate constant | |
| $K_3$ | $0.4\,s^{-1}$ | Cross-bridge formation rate constant | |
| $K_4$ | $0.1\,s^{-1}$ | Cross-bridge detachment rate constant | |
| $K_5$ | $0.5\,s^{-1}$ | Myosin dephosphorylation rate constant | |
| $K_6$ | $\gamma_{cross}[Ca^{2+}]_{cyt}^3$ | Rate in changing from AM to AMp | |
| $K_7$ | $0.1\,s^{-1}$ | Latch state to free myosin detachment rate | |
| IP$_3$ | | | |
| $\eta_{IP_3}$ | $11.725\,(\mu M \cdot s)^{-1}$ | Effective signal gain parameter | [57] |
| $k_{deg,IP_3}$ | $1.25\,s^{-1}$ | IP$_3$ degradation rate | |
| $k_c$ | $0.4\,\mu M$ | Dissociation constant for the $Ca^{2+}$ binding site on the PLC molecule | |
| $PIP_{2rr}$ | $0.1\,s^{-1}$ | PIP 2 replenishment rate | |
| $[PIP_2]_{tot}$ | $118.61\,\mu M$ | Total number of $PIP_2$ molecules | [55] |

The primary phenotype of interest that is caused by perturbed calcium dynamics is vaso-constriction. We model vasoconstriction and dilation indirectly through a four-state model of calcium-dependent actin/myosin dynamics, free cross-bridges (M), phosphorylated cross-bridges (Mp), attached phosphorylated cross-bridges (AMp) and attached dephosphorylated non-cycling cross bridge (AM) [54]

$$
\begin{aligned}
\mathrm{M} &= 1 - \mathrm{Mp} - \mathrm{AMp} - \mathrm{AM} \\
\frac{\mathrm{dMp}}{\mathrm{d}t} &= K_4 \times \mathrm{AM}p + K_1 \times \mathrm{M} - (K_2 + K_3) \times \mathrm{Mp} \\
\frac{\mathrm{dAMp}}{\mathrm{d}t} &= K_3 \times \mathrm{M}p + K_6 \times \mathrm{AM} - (K_4 + K_5) \times \mathrm{AMp} \\
\frac{\mathrm{dAM}}{\mathrm{d}t} &= K_5 \times \mathrm{AMp} - (K_7 + K_6) \times \mathrm{AM},
\end{aligned}
\tag{5}
$$

with fixed parameters in Table 1.

The tension generated by the smooth muscle cell is a linear function

$$
F_r \propto (\mathrm{AMp} + \mathrm{AM})
$$

of the total associated actin-associated myosin. Since three calcium ions are effectively buffered when myosin is in either the Mp or AMp states, myosin binding induces a change in cytosolic free calcium given by the reaction flux

$$
J_{\mathrm{cyt} \to \mathrm{cyt}} = -[\mathrm{Myo}]\frac{\mathrm{d}}{\mathrm{d}t}(\mathrm{Mp} + \mathrm{AMp}).
\tag{6}
$$

The disassociation of myosin from actin is energy dependent. For the sake of simplicity, we assume that ATP production meets the needs of this system.

The remaining contributors to cytosolic calcium changes lie in the specific mechanisms through which the cytosol interacts with the other compartments. Inositol trisphosphate, ($\mathrm{IP}_3$), is an important modulator of these interactions. We model the dynamics of of $\mathrm{IP}_3$ using the model of [57], where

$$
\frac{\mathrm{d}[\mathrm{IP}_3]}{\mathrm{d}t} = r_h[\mathrm{PIP}_2] - k_{\mathrm{deg},\mathrm{IP}_3}[\mathrm{IP}_3],
\tag{7}
$$

which is coupled to the dynamics for phosphatidylinositol (4,5)-bisphosphate ($\mathrm{PIP}_2$)

$$
\frac{\mathrm{d}[\mathrm{PIP}_2]}{\mathrm{d}t} = -(r_h + \mathrm{PIP}_{2\mathrm{rr}}) \cdot [\mathrm{PIP}_2] + \mathrm{PIP}_{2\mathrm{rr}} \cdot ([\mathrm{PIP}_2]_{\mathrm{tot}} - [\mathrm{IP}_3]).
\tag{8}
$$

Note that we only model $\mathrm{IP}_3$ and $\mathrm{PIP}_2$ in the cytosolic compartment—for this reason we drop the cytosolic subscript. Both Eqs 7 and 8 are sensitive to cytosolic calcium concentration due to the relationship

$$
r_h = \eta_{\mathrm{IP}_3}\frac{[\mathrm{Ca}^{2+}]_{\mathrm{cyt}}}{k_c + [\mathrm{Ca}^{2+}]_{\mathrm{cyt}}}[\mathrm{G}],
\tag{9}
$$

where the concentration of G-protein $[\mathrm{G}] = 3.314 \times 10^{-5}\,\mu\mathrm{M}$ is assumed to be fixed.

## Mitochondria

In the mitochondria, we assume that there is an excess of calcium and that calcium-phosphate clusters have formed. Additionally, we assume that this process has come to equilibrium so that calcium and phosphate are balanced at their saturation concentrations. By conservation,

the mitochondrial calcium concentration obeys

$$\frac{d[Ca^{2+}]_{mit}}{dt} = J_{mit \to mit} - J_{mit \to cyt},$$

(10)

where $J_{mit \to mit}$ accounts for the exchange of calcium between the free solution and clusters. We make the assumption that this exchange is rapid, relative to exchange with the cytosol and it could compensate the concentration change induced by $J_{mit \to cyt}$. These assumptions imply that $[Ca^{2+}]_{mit}$ is a constant so long as clusters are not depleted and the composition of the solution does not change drastically. For these conditions to hold, we make the additional assumptions that the pH and concentration of phosphate are held nearly constant by other processes which we do not explicitly model.

Excess mitochondrial calcium is the driver for perturbations in calcium dynamics in our model, where we include three explicit mechanisms for calcium exchange between mitochondria and the cytosol. Although the free ionic calcium concentration within the mitochondria is fixed, due to supersaturation, the net amount of calcium present within the mitochondria varies as a result of exchange with the cytosol. In normal physiology, mitochondria are calcium buffers of last resort [51, 58, 59], taking in calcium through a combination of the mitochondrial calcium uniporter (MCU) and the mitochondrial sodium calcium exchanger (NCLX). Parameter values particular to the mitochondria are in Table 2.

We assume that the flux through the uniporter is unidirectional (from cytosol to mitochondria), with rate given by [56]

$$J_{mcu} = -v_{mcu} \frac{\frac{[Ca^{2+}]_{cyt}}{K_{mcu,1}} \left(1 + \frac{[Ca^{2+}]_{cyt}}{K_{mcu,1}}\right)^3}{\left(1 + \frac{[Ca^{2+}]_{cyt}}{K_{mcu,1}}\right)^4 + \frac{L_{mcu}}{\left(1 + \frac{[Ca^{2+}]_{cyt}}{K_{mcu,2}}\right)^{2.3}}} e^{p_1 \Delta \Psi_m}.$$

(11)

The NCLX is a bidirectional $Na^+/Ca^{2+}$ exchanger [60] for $[Ca^{2+}]$ extrusion out of mitocondria. We model its calcium flux only from mitochondrial to cytosol through the expression [56]

$$J_{nclx} = v_{nclx} \left(\frac{[Ca^{2+}]_{mit}}{[Ca^{2+}]_{cyt}}\right) e^{p_2 \Delta \Psi_m},$$

(12)

where the mitochondrial membrane potential $\Delta \Psi_m$ is considered fixed. Finally, we model an Ohmic leak, which is bidirectional depending on the compete between membrane potential

**Table 2. Mitochondria model parameters.** MCU parameters taken from [56] and adjusted based on [33] Figure 2D.

| Parameter | Value | Description | Notes |
|---|---|---|---|
| $v_{mcu}$ | $4.4 \times 10^{-6}\ \mu M/s$ | Rate constant of the MCU | this work |
| $v_{nclx}$ | $0.13\ \mu M/s$ | Rate constant of the NCLX | |
| $k_c$ | $0.4\ \mu M$ | Dissociation constant for calcium binding to PLC | [56] |
| $\Delta \Psi_m$ | $140\ mV$ | Mitochrodral membrane potential | |
| $p_1$ | $0.1\ mV^{-1}$ | Voltage dependence coefficient of MCU activity | |
| $p_2$ | $0.0161\ mV^{-1}$ | Voltage dependence coefficient of NCLX activity | |
| $K_{mcu,1}$ | $6\ \mu M$ | Dissociation constant for $Ca^{2+}$ translocation by the MCU | |
| $K_{mcu,2}$ | $0.38\ \mu M$ | Dissociation constant for MCU activation by $Ca^{2+}$ | |
| $L_{mcu}$ | 50 | Allosteric equilibrium constant for uniporter conformations | |

and calcium Nernst potential, through non-specific mechanisms as

$$J_{\mathrm{mit,leak}} = v_{\mathrm{mit,leak}}(E_{\mathrm{Ca,mit}} - \Delta\Psi_m), \tag{13}$$

where the mitochondrial calcium Nernst potential is

$$E_{\mathrm{Ca,mit}} = \frac{RT}{2F}\log\left(\frac{[\mathrm{Ca}^{2+}]_{\mathrm{mit}}}{[\mathrm{Ca}^{2+}]_{\mathrm{cyt}}}\right). \tag{14}$$

The conductance of the leak current is set so that the mitochondrial flux is zero when $[\mathrm{Ca}^{2+}]_{\mathrm{mit}}$ = $[\mathrm{Ca}^{2+}]_{\mathrm{cyt}}$ = 0.1 $\mu$M. We express the net flux between the mitochondrial compartment and cytosol as a sum over the three contributors,

$$J_{\mathrm{mit}\to\mathrm{cyt}} = J_{\mathrm{mcu}} + J_{\mathrm{nclx}} + J_{\mathrm{mit,leak}}. \tag{15}$$

## Endoplasmic reticulum

Via exchange with the cytosol, the ER calcium concentration follows

$$\frac{\mathrm{d}[\mathrm{Ca}^{2+}]_{\mathrm{er}}}{\mathrm{d}t} = -\frac{r_{\mathrm{er}}^{\mathrm{cyt}}}{b_{\mathrm{er}}}J_{\mathrm{er}\to\mathrm{cyt}} \tag{16}$$

with dynamic buffering given

$$b_{\mathrm{er}} = 1 + \frac{K_{\mathrm{CalrC}}[\mathrm{CalrC}]_{\mathrm{er}}}{(K_{\mathrm{CalrC}} + [\mathrm{Ca}^{2+}]_{\mathrm{er}})^2} + \frac{K_{\mathrm{CalrP}}[\mathrm{CalrP}]_{\mathrm{er}}}{(K_{\mathrm{CalrP}} + [\mathrm{Ca}^{2+}]_{\mathrm{er}})^2}. \tag{17}$$

where all quantities are fixed and listed in Table 3.

Within the ER, we model the interplay between calcium intake through the ubiquitous sarco/endo-plasmic reticulum (SERCA) buffering protein and calcium-induced-calcium-release (CICR) through the calcium-triggered ryanodine receptors (RyR) and IP$_3$-sensitive receptors (IP$_3$Rs). Hence, we write

$$J_{\mathrm{er}\to\mathrm{cyt}} = J_{\mathrm{serca}} + J_{\mathrm{ryr}} + J_{\mathrm{ipr}}. \tag{18}$$

**Sarco/endoplasmic reticulum Ca$^{2+}$-ATPase.** The ER, through the SERCA pump protein, is the primary cellular calcium buffer under physiological conditions. SERCA hydrolyzes ATP to pump calcium against its gradient into the ER, where free ionic calcium is two orders of magnitude more concentrated than in the cytosol. We model the action of SERCA using the bufferless approximation of [61],

$$J_{\mathrm{serca}} = \frac{2(-K_1^2 K_3^2 k_{-2} k_{-4}[\mathrm{Ca}^{2+}]_{\mathrm{er}}^2 + k_2 k_4[\mathrm{Ca}^{2+}]_{\mathrm{cyt}}^2)[\mathrm{SERCA}]}{[\mathrm{Ca}^{2+}]_{\mathrm{er}}^2[\mathrm{Ca}^{2+}]_{\mathrm{cyt}}^2 K_3^2(k_2 + k_{-2}) + [\mathrm{Ca}^{2+}]_{\mathrm{cyt}}^2(k_4 + k_2) + [\mathrm{Ca}^{2+}]_{\mathrm{er}}^2 K_1^2 K_3^2(k_{-2} + k_{-4}) + K_1^2(k_4 + k_{-4})}, \tag{19}$$

where the model parameters are given in Table 3.

**IP$_3$ receptors.** The open probability of IP$_3$R channel is regulated by cytosolic calcium concentration and IP$_3$ concentration. For determining flux through IP$_3$R, the four-state channel model of [62] is used to find the open probability of IP$_3$R. Let $X_{ij}$ i, j = 0, 1, be the probability of

**Table 3. ER model parameters.**

| Parameter | Value | Description | Notes |
|---|---|---|---|
| Buffer | | | |
| $K_{CalC}$ | $2 \times 10^3 \, \mu M$ | Dissociation rate of Site C | this work |
| $[CalC]_{er}$ | $7.2 \times 10^3 \, \mu M$ | Concentration of site C | this work |
| $K_{CalP}$ | $10 \, \mu M$ | Dissociation rate of Site P | this work |
| $[CalP]_{er}$ | $7.2 \times 10^2 \, \mu M$ | Concentration of site P | this work |
| SERCA | | | |
| $K_1$ | $\sqrt{0.7} \, \mu M$ | SERCA in cytosol equilibrium constant ratio | [61] |
| $K_3$ | $\sqrt{1.111111 \times 10^{-5}} \, \mu M$ | SERCA in ER equilibrium constant ratio | [61] |
| $k_2$ | $0.6 \times \xi \, s^{-1}$ | SERCA pump rate constants | [61] |
| $k_{-2}$ | $0.97 \times \xi \, s^{-1}$ | SERCA pump rate constants | [61] |
| $k_4$ | $0.4 \times \xi \, s^{-1}$ | SERCA pump rate constants | [61] |
| $k_{-4}$ | $1.2 \times 10^{-3} \times \xi \, s^{-1}$ | SERCA pump rate constants | [61] |
| [SERCA] | $182 \, \mu M$ | Concentration of pump protein | this work |
| $\xi$ | 10 | Pump speed ratio | this work |
| IP$_3$R | | | |
| $v_{ipr}$ | $1000 \, s^{-1}$ | Rate constant of the IP$_3$R | this work |
| $a_1$ | $167.6 \, (\mu M)^{-1} \, s^{-1}$ | IP$_3$ Receptor binding constants | [62] |
| $a_2$ | $3.81 \, (\mu M)^{-1} \, s^{-1}$ | Calcium inhibition constant for binding | [62] |
| $a_3$ | $413.4 \, (\mu M)^{-1} \, s^{-1}$ | IP$_3$ receptor binding constants | [62] |
| $a_4$ | $0.3101 \, (\mu M)^{-1} \, s^{-1}$ | Calcium inhibition constant for binding | [62] |
| $a_5$ | $53.9 \, (\mu M)^{-1} \, s^{-1}$ | Calcium activation constant for binding | [62] |
| $b_1$ | $228 \, s^{-1}$ | IP$_3$ receptor dissociation constants | [62] |
| $b_2$ | $0.409 \, s^{-1}$ | Calcium inhibition constant for dissociation | [62] |
| $b_3$ | $188.5 \, s^{-1}$ | IP$_3$ receptor dissociation constants | [62] |
| $b_4$ | $0.096 \, s^{-1}$ | Calcium inhibition constant for dissociation | [62] |
| $b_5$ | $4.52 \, s^{-1}$ | Calcium activation constant for dissociation | [62] |
| RyR | | | |
| $v_{ryr}$ | $0.2 \, s^{-1}$ | Rate constant of the RyR | this work |
| $K_{r1}$ | $2.5 \, (\mu M)^{-2} \, s^{-1}$ | Activation rate constant | [54] |
| $K_{r2}$ | $1.5 \, (\mu M)^{-1} \, s^{-1}$ | Inactivation rate constant | [54] |
| $K_{-r1}$ | $7.6 \, s^{-1}$ | Unbinding rate constant from activation | [54] |
| $K_{-r2}$ | $84 \, s^{-1}$ | Unbinding rate constant from inactivation | [54] |

IP$_3$R in the corresponding state indexed $(i, j)$ and modeled by the dynamical system

$$
\begin{pmatrix} \dot{X}_{00} \\ \dot{X}_{01} \\ \dot{X}_{10} \end{pmatrix} = \begin{pmatrix} 0 \\ b_5 \\ b_2 \end{pmatrix}
+ \begin{bmatrix} -\dfrac{(a_4 k_1 + a_5 + a_2)[Ca^{2+}]_{cyt}}{1 + k_1} & \dfrac{b_4 k_3 + b_2}{1 + k_3} & b_5 \\[2ex] \dfrac{(a_4 k_1 + a_2)[Ca^{2+}]_{cyt}}{1 + k_1} - b_5 & -\dfrac{b_4 k_3 + b_2 + a_5[Ca^{2+}]_{cyt}}{1 + k_3} - b_5 & -b_5 \\[2ex] \dfrac{a_5[Ca^{2+}]_{cyt}}{1 + k_1} - b_2 & -b_2 & -a_2[Ca^{2+}]_{cyt} - b_5 - b_2 \end{bmatrix} \begin{pmatrix} X_{00} \\ X_{01} \\ X_{10} \end{pmatrix} \quad (20)
$$

where $k_i = \frac{b_i}{a_i[\text{IP}_3]}$, $i = 1, 3$. The fourth state has been eliminated using the conservation condition $X_{10} = 1 - X_{00} - X_{01} - X_{11}$.

The whole-cell IP$_3$R flux, which is from ER to cytosol, is given by

$$J_{\text{ipr}} = v_{\text{ipr}}(X_{10}^4 + 4X_{10}^3(1 - X_{10}))([\text{Ca}^{2+}]_{\text{er}} - [\text{Ca}^{2+}]_{\text{cyt}}), \tag{21}$$

where $(X_{10}^4 + 4X_{10}^3(1 - X_{10}))$ is the signal channel open probability and the model parameters are given in Table 3.

**Ryanodine receptors.** For the RyR, a four-state model developed by Yang et al [54] is used. Those four states are free receptors $R_{00}$, receptors with [Ca$^{2+}$] bound to activation sites $R_{01}$, receptors with [Ca$^{2+}$] bound to inactivation sites $R_{01}$, and receptors with both activation and inactivation sites bound by [Ca$^{2+}$], $R_{11}$.

$$
\begin{pmatrix} \dot{R}_{10} \\ \dot{R}_{11} \\ \dot{R}_{01} \end{pmatrix} = \begin{pmatrix} K_{r1}[\text{Ca}^{2+}]_{\text{cyt}}^2 \\ 0 \\ K_{r2}[\text{Ca}^{2+}]_{\text{cyt}} \end{pmatrix}
$$

$$
+ \begin{bmatrix} -(K_{-r1} + K_{r2}[\text{Ca}^{2+}]_{\text{cyt}}) - K_{r1}[\text{Ca}^{2+}]_{\text{cyt}}^2 & K_{-r2} - K_{r1}[\text{Ca}^{2+}]_{\text{cyt}}^2 & -K_{r1}[\text{Ca}^{2+}]_{\text{cyt}}^2 \\ K_{r2}[\text{Ca}^{2+}]_{\text{cyt}} & -K_{-r1} - K_{-r2} & K_{r1}[\text{Ca}^{2+}]_{\text{cyt}}^2 \\ -K_{r2}[\text{Ca}^{2+}]_{\text{cyt}} & K_{-r1} - K_{r2}[\text{Ca}^{2+}]_{\text{cyt}} & K_{-r2} - K_{r1}[\text{Ca}^{2+}]_{\text{cyt}}^2 - K_{r2}[\text{Ca}^{2+}]_{\text{cyt}} \end{bmatrix} \tag{22}
$$

$$
\times \begin{pmatrix} R_{10} \\ R_{11} \\ R_{01} \end{pmatrix},
$$

and the correponding flux induced by RyR channel,which is also from ER to cytosol, is given by

$$J_{\text{ryr}} = v_{\text{ryr}}R_{10}^2([\text{Ca}^{2+}]_{\text{er}} - [\text{Ca}^{2+}]_{\text{cyt}}), \tag{23}$$

where the model parameters are given in Table 3.

## Plasma membrane

The interactions between the cytosol of the smooth muscle cell and the extracellular space that we consider are the calcium-dependent ones of [55], though the sodium-calcium exchanger (NCX) is taken from [63]. The NCX channel, which moves calcium out of cytosol to extracelluar space, follows

$$J_{\text{ecs,ncx}} = \frac{k_{\text{ncx2}}}{2FV_{\text{cyt}}} \frac{([\text{Na}^+]_{\text{cyt}}^3[\text{Ca}^{2+}]_{\text{ecs}}\phi_F - [\text{Na}^+]_{\text{ecs}}^3[\text{Ca}^{2+}]_{\text{cyt}}\phi_R)}{(1 + d_{\text{ncx}}([\text{Ca}^{2+}]_{\text{cyt}}[\text{Na}^+]_{\text{ecs}}^3 + [\text{Ca}^{2+}]_{\text{ecs}}[\text{Na}^+]_{\text{cyt}}^3))} \frac{1}{1 + \left(\frac{k_{\text{ncx1}}}{[\text{Ca}^{2+}]_{\text{cyt}}}\right)^2}. \tag{24}$$

The Voltage-gated calcium channel (VOCC), which is responsible for the influx of calcium into the cytosol from extracelluar space, follows

$$J_{\text{vocc}} = Q_{\text{vocc}}\bar{d}_l\bar{f}_l(E_{\text{Ca,ecs}} - \phi_{\text{ecs}}), \tag{25}$$

where

$$E_{\text{Ca,ecs}} = \frac{RT}{2F} \log\left(\frac{[\text{Ca}^{2+}]_{\text{ecs}}}{[\text{Ca}^{2+}]_{\text{cyt}}}\right), \tag{26}$$

$$\phi_F = \exp\left(\eta_{\text{ncx}}\phi_{\text{ecs}}\frac{F}{RT}\right), \quad \phi_R = \exp\left((\eta_{\text{ncx}} - 1)\phi_{\text{ecs}}\frac{F}{RT}\right), \tag{27}$$

$$\bar{d}_l = \frac{1}{\left(1 + \exp\left(-\frac{\phi_{\text{ecs}}}{8.3 \text{ mV}}\right)\right)}, \quad \text{and} \quad \bar{f}_l = \frac{1}{\left(1 + \exp\left(\frac{(\phi_{\text{ecs}} + 42 \text{ mV})}{9.1 \text{ mV}}\right)\right)}. \tag{28}$$

The non-specific leak and (Plasma membrane Ca2+ ATPase) PMCA follow

$$J_{\text{ecs,leak}} = -g_{\text{leak,ecs}}(\phi_{\text{ecs}} - E_{\text{Ca,ecs}}) \tag{29}$$

$$J_{\text{pmca}} = -Q_{\text{pmca}}\frac{[\text{Ca}^{2+}]_{\text{cyt}}}{[\text{Ca}^{2+}]_{\text{cyt}} + k_{\text{pmca}}}. \tag{30}$$

Overall,

$$J_{\text{ecs}\rightarrow\text{cyt}} = J_{\text{ecs,leak}} + J_{\text{pmca}} + J_{\text{ecs,ncx}} + J_{\text{vocc}}. \tag{31}$$

The model parameters are given in Table 4.

## Tracking calcium clearance

We assume that the extracellular space is a bulk bath, thereby assuming that $[\text{Ca}^{2+}]_{\text{ecs}}$ = 1.3 mM. Yet, while the ECS is assumed to have a constant concentration, we still track the net calcium flow from the cytosol to the ECS, through mass balance, using the differential equation

$$\frac{d[\text{Ca}^{2+}]_{\text{sink}}}{dt} = -J_{\text{ecs}\rightarrow\text{cyt}}. \tag{32}$$

**Table 4. Plasma-membrane and extracellular specific model parameters.**

| Parameter | Value | Description | Notes |
|-----------|-------|-------------|-------|
| $\phi_{\text{ecs}}$ | −54 mV | Membrane potential | [55] |
| $V_{\text{cyt}}$ | 0.7 pL | Cytosol volume | |
| $\eta_{\text{ecs}}$ | 0.35 | Position of energy barrier of the NCX | |
| $[\text{Na}^+]_{\text{ecs}}$ | 140 mM | Extracellular $[\text{Na}^+]$ concentration | |
| $[\text{Na}^+]_{\text{cyt}}$ | 8 mM | Cytosolic $[\text{Na}^+]$ concentration | |
| $d_{\text{ncx}}$ | $(0.01\ \mu\text{M})^{-4}$ | Scaling factor for the NCX | [63] |
| $k_{\text{ncx1}}$ | $0.125\ \mu\text{M}$ | Scaling factor for the NCX | |
| $Q_{\text{pmca}}$ | $0.04\ \mu\text{M s}^{-1}$ | Rate constant of the PMCA | this work |
| $k_{\text{ncx2}}$ | $0.5\ \mu\text{A}$ | Scaling factor for the NCX | |
| $g_{\text{leak,ecs}}$ | $3.0 \times 10^{-5}\ \mu\text{M (mV} \cdot s)^{-1}$ | Rate constant of the leak | |
| $k_{\text{pmca}}$ | $0.15\ \mu\text{M}$ | Disociation constant of the PMCA | |
| $Q_{\text{vocc}}$ | $0.10\ \mu\text{M (mV} \cdot s)^{-1}$ | Rate constant of the VOCC | |
| $F$ | $96487\ \text{C} \cdot \text{mol}^{-1}$ | Faraday constant | |

Similarly, we assume that the calcium concentration is fixed at an elevated state determined by equilibrium solubility of calcium and phosphate, under the assumption that both species are in excess and that the pH is stable. However, we track the net release of calcium from the mitochondria through the differential equation

$$\frac{d[Ca^{2+}]_{source}}{dt} = J_{mit \to cyt}. \tag{33}$$

The net rate of calcium release from the cell is the primary quantity of interest in our model, as it provides a picture of the overall timescale of perturbed calcium dynamics.

The leakage of calcium from mitochondria is what drives the dynamics in our model. Due to stoichiometric considerations, leakage corresponds to calcium dissolution which also affects the pH within the mitochondria. Due to this fact, the leakage of calcium from the mitochondria also implies a driving source of ATP production.

### Contribution of dissolution to ATP production

Aerobic ATP production is driven by energy from the proton gradient between the mitochondrial matrix and the mitochondrial inner membrane. In aerobic conditions, this gradient is maintained by proton pumps in the electron transport chain [64]. Protons within the inner mitochondrial membrane are in passive exchange with the bulk cytosol. The pumps move protons from the matrix into the inner membrane. They are extremely efficient; on average only a small number of protons are present within each mitochondrial matrix, maintaining a stable alkaline environment of pH ≈ 7.8 [65]. Hence, the main determinant of the gradient is the absence or presence of protons in the matrix.

The mitochondrial $F_0F_1$-ATP synthase harnesses the passive flux of protons through its inner pore, in order to produce ATP from ADP and free inorganic phosphate. Being reversible, this mechanism depends crucially on maintenance of the gradient, typically by oxidative phosphorylation. Any process that removes protons from the mitochondrial matrix helps to sustain ATP production, where an estimated three protons contributes to the production of an ATP molecule [66].

Liberation of calcium also results in liberation of free inorganic phosphate and hydroxide ions, both of which associate with protons. Hence, calcium phosphate dissolution can have an energetically protective effect by sustaining aerobic respiration in anaerobic settings. We quantify the resulting rate of ATP production through this mechanism by noting that the production rate is proportional to $J_{mit \to cyt}$, under the assumption of quasi-steady equilibrium in the interplay between dissolution and aggregation.

### Steady-state parameterization

Our mathematical model has several free parameters. By enforcing the steady-state of cell under normal circumstances as shown in Table 5, we constrain the dimensionlity of these parameters.

### Results

We implemented our model in Julia 1.1 using `DifferentialEquations.jl` [68], with source code available at github:xsxztr/xsxztr-Post-CSD-code-with-Joshua-C.-Chang. We used the default stiff ODE solver (called with `alg_hints = [:stiff]`), which was sufficiently stable at automatic step sizes to yield solutions at the default relative tolerance of $10^{-6}$. In each of our simulations, all of the state variables in our model are initialized to satisfy steady-state concentrations for free calcium. We present results relative to a baseline parameter regime

**Table 5. Steady state free ionic calcium concentrations enforced in all of our model simulations.**

| State | Value | Notes |
|---|---|---|
| $[Ca^{2+}]_{cyt}$ | 0.1 $\mu$M | [67] |
| $[Ca^{2+}]_{ecs}$ | 1300 $\mu$M | [30] |
| $[Ca^{2+}]_{mit}$ | 0.1 $\mu$M | [56] |
| $[Ca^{2+}]_{er}$ | 500 $\mu$M | this work |
| $R_{01}$ | 0.001775 | [54] |
| $R_{10}$ | 0.003272 | |
| $R_{11}$ | 5.8440e-6 | |
| $X_{01}$ | 0.2430 | |
| $X_{10}$ | 0.004820 | |
| $X_{00}$ | 0.7475 | |

with behavior matching the vascular dynamics seen in [12]. We ascertained the frequency of any oscillations using DSP.jl, (github: JuliaDSP/DSP.jl) which is a common digital signal processing package for Periodogram estimation.

## Effect of elevations in $[Ca^{2+}]_{mit}$

We first fixed the model parameters to the reference values in Tables 1, 4 and 5; the only free parameter was the free ionic calcium concentration within the mitochondria. In these simulations the stabilizing leak fluxes are calibrated in order to maintain the original unperturbed steady state.

Fig 2 shows the bifurcation from steady state to oscillatory cytosolic calcium dynamics above a critical threshold for $[Ca^{2+}]_{mit} \approx 0.245$ $\mu$M. The magnitude of these oscillations increases with the mitochondrial calcium concentration. The same bifurcation is present in the other model variables, for instance, the fraction of myosin light chain in the force-generating state ($F_r$). Although the net movement of calcium is from the mitochondria, to the cytosol, to the extracellular space, the direction of this flow oscillates as well.

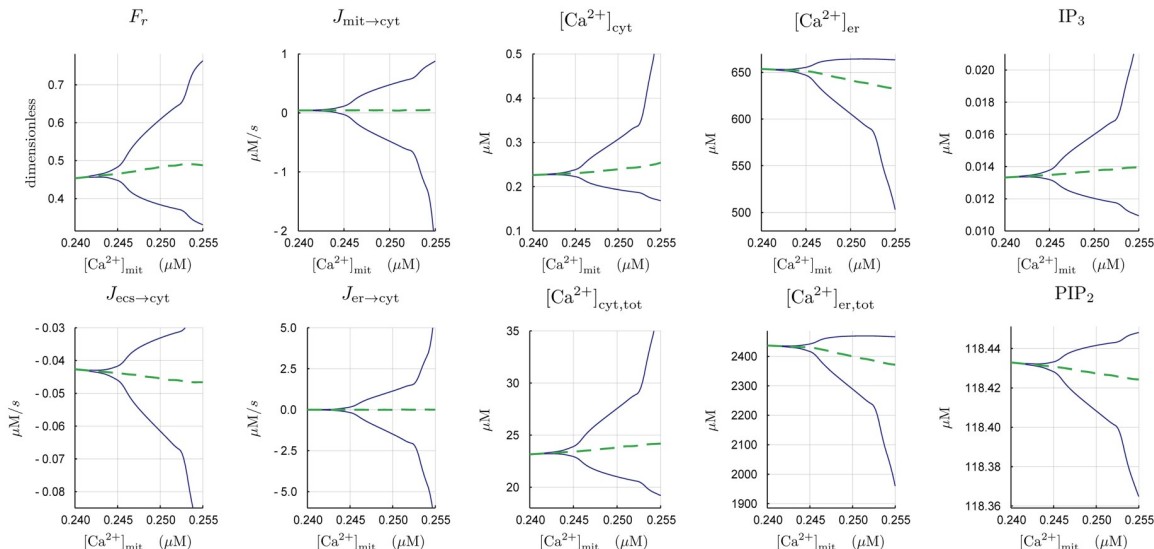

**Fig 2. Hopf bifurcation diagrams as a function of $[Ca^{2+}]_{mit}$.** A supercritical Hopf bifurcation initiates at around $[Ca^{2+}]_{mit} \approx 0.245$ $\mu$M. The period-averaged value for each variable is shown in dashed-green.

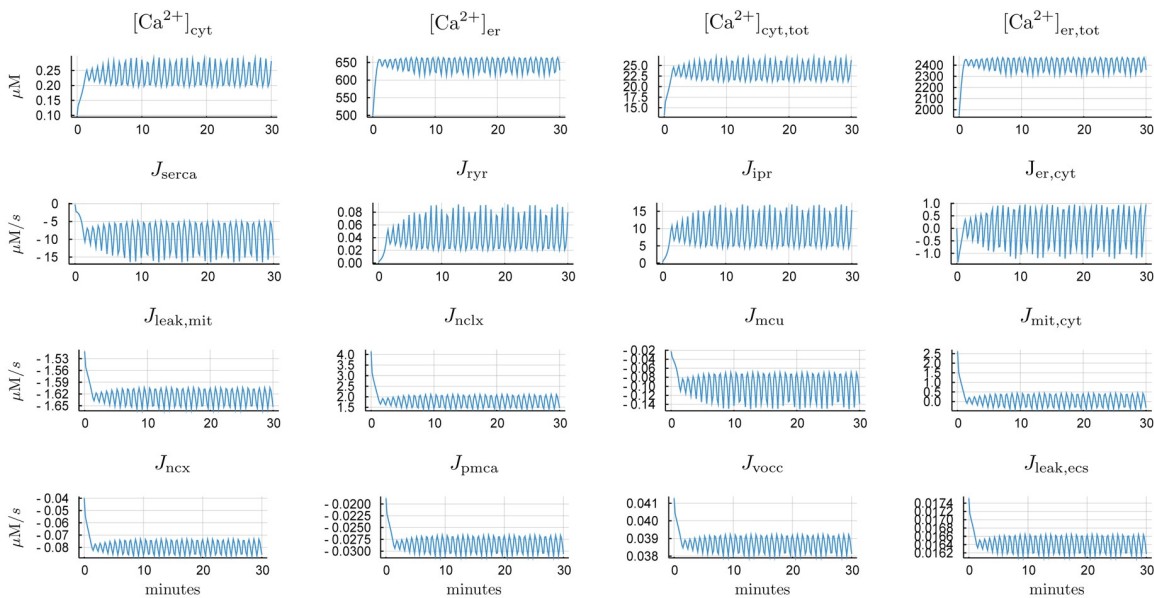

**Fig 3. Model time courses for $[Ca^{2+}]_{mit} = 0.25\,\mu M$.** Positive fluxes correspond to flow into the cytosol. We distinguish between free and total calcium in each of the ER and cytosol. Mitochondrial calcium assumed to be in quasi-steady equilibrium with respect to dissolution and aggregation.

Fig 3 presents time courses of $[Ca^{2+}]_{mit} = 0.25\,\mu M$. A transient period of approximately three minutes characterizes these simulations, where calcium from the mitochondria loads into the cytosol, primarily into the endoplasmic reticulum as shown in both the free and total calcium concentrations. The loading is primarily through action of SERCA, and is counteracted by calcium induced calcium release (CICR) through the $IP_3R$ and ryanodine receptor channels.

The dynamics in this set of simulations are consistent with the observed vasoactivity of Figure 4A in [12], where a similar trace is reproduced in Fig 4D. Fig 4 represents the macroscale physiological implications of the cellular calcium dynamics from Fig 3. In half an hour, approximately 80 $\mu M$ of free calcium is expelled from the cell ($[Ca^{2+}]_{sink}$), where the concentration is relative to the cytosolic volume. More calcium leaves the mitochondria ($[Ca^{2+}]_{source}$), with the difference accounted for by the combination of cytosolic and cellular buffers. Together, these two variables determine the timescale over which excess calcium in the mitochondrial matrix is depleted. Concomitant with the calcium dynamics, oscillatory vascular dynamics within a range of constrictions is predicted.

We study the parametric determinants of calcium dynamics in our model by looking at the impact of relevant fluxes on the time scale of calcium fluxes with fixed elevated mitochondrial concentration 0.25 $\mu M$ via

$$J_{(\cdot)\to cyt} \to \alpha_{(\cdot)} J_{(\cdot)\to cyt}. \tag{34}$$

where $\alpha_{(\cdot)}$ is the timescale factor which mean how fast of each compartment achieves steady states. Note that these transformations preserve steady state in our model because the model is calibrated to zero flux at the prescribed steady state described in the aforementioned tables. Fig 5 shows how rescalings of the three fluxes, for $\alpha_{(\cdot)} \in [0.1, 10]$, affect the presence of stable oscillations in cytosolic calcium concentration. From these simulations, it is clear that the endoplasmic reticulum is crucial for the development of oscillations. There were no oscillations observed for $\log_{10}\alpha_{er} \lesssim -0.5$.

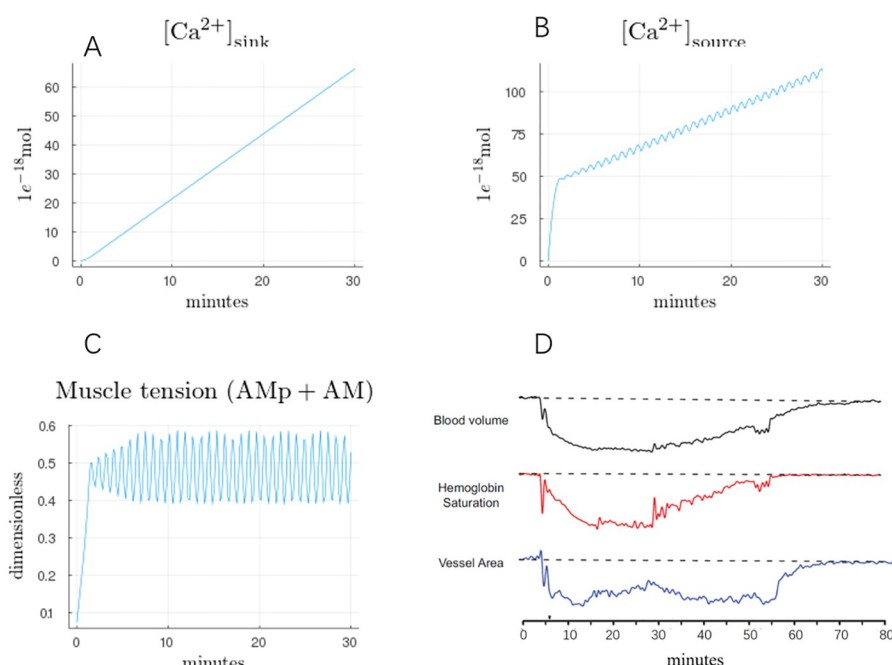

**Fig 4. Cumulative calcium movement and vascular activity as a consequence of elevated mitochondrial calcium** ([Ca$^{2+}$]$_{mit}$ = 0.25 $\mu$M). A) Sink refers to the cumulative calcium exiting the cell and entering the ECS. B) Source refers to the cumulative calcium exiting the mitochondrial matrix. All concentrations relative to the cytoplasmic volume which is assumed to be 0.7 pL. C) Muscle tension predicted by our model. D) **Trace similar to Fig.4A in** [12], showing in-vivo vascular dynamics following CSD.

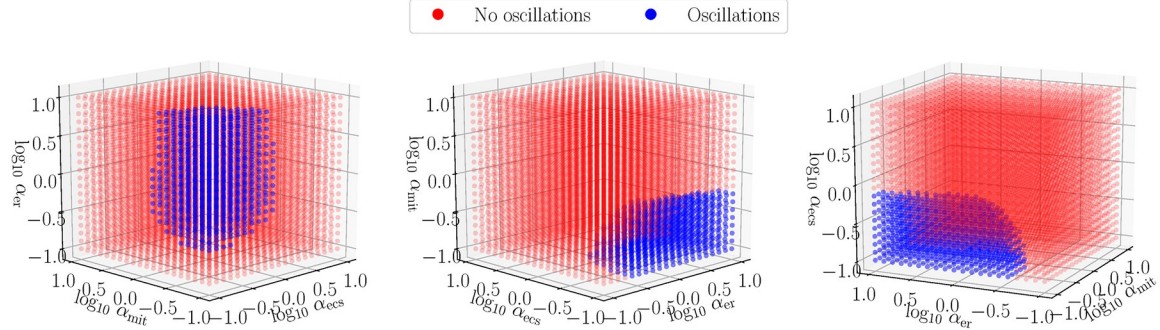

**Fig 5. Presence of oscillations as a function of flux rescalings defined in Eq 34, rotations of axes shown for visability.**

## Determinants of timescale

As shown in Fig 2, elevations in [Ca$^{2+}$]$_{mit}$ drive changes in calcium dynamics. The persistence of these perturbations depends on the timescale over which calcium remains elevated in the mitochondria, which is controlled by the rate of calcium extrusion. Eqs 32 and 33 describe the cumulative extrusion of calcium from the mitochondria and the cell respectively, as shown in Fig 4.

In Fig 6, for $\log_{10} \alpha_{(\cdot)} \in [-1, 1]$, we present the effect of the flux rescalings on the overall rate of calcium extrusion from the cell, conditional on the presence or absence of oscillations. It is evident that the presence of oscillations does not impact the rate of efflux—as shown in the first

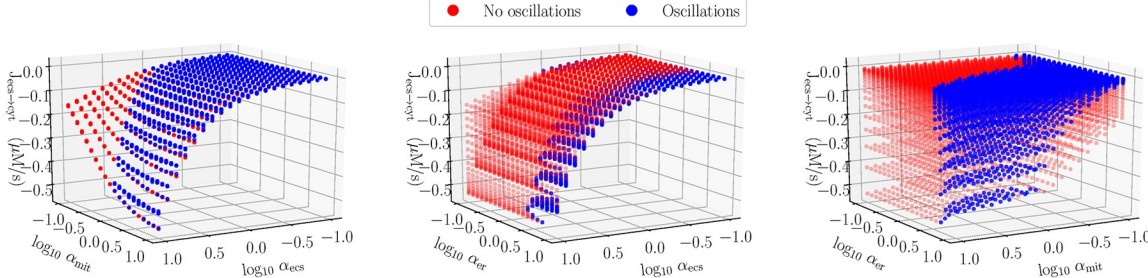

**Fig 6. The net period-averaged calcium flux between the extracellular space and cytosol as the channel and pump rates are rescaled according to Eq 34.** Left to right: flux as a function of mitochondrial and plasma membrane rescaling, as a function of ER and plasma membrane rescaling, and as a function of ER and mitochondrial rescaling. Blue denotes the presence of stable oscillations.

panel, $J_{ecs \to cyt}$ is largely independent of $\alpha_{er}$ as long as the two other rescalings are taken into account. The magnitude of the flux also increases sub-linearly with respect to increase of the mitochondrial fluxes though nearly linearly with respect to plasma membrane rescaling. The kinetics of the plasma membrane mechanics are the main determinant of net extrusion rate.

At rest, the extracellular space comprises approximately 20% of the cerebral tissue volume [69, 70], with a resting ionic calcium concentration of approximately 1.3 mM. At the peak of CSD, the extracellular calcium concentration drops to under 0.1 mM while its volume fraction drops by half. To understand the timescale implied in the simulations of Figs 3 and 6, suppose that there is only one cell type within the tissue, that comprises the remaining 80% of the volume, and absorbs all of the calcium from the ECS during CSD. If we assume conservatively that, due to the volume ratios, calcium dilutes by a factor of four in the cytosol, an effective concentration of over 300 $\mu$M enters into these cells. However, the cells would not experience elevations of free calcium near the level in the cytosol since the calcium would be buffered. If the mitochondria absorb approximately a third of this excess calcium, then the simulation of Fig 3 predicts a timescale of over 30 minutes to expel the absorbed calcium.

In all of these simulations, the direction of the net calcium flux at 30 minutes is out of the cell. However, as shown in Fig 7, there are a minority of simulations whose parameter sets yielded a net inward mitochondrial flux at 30 minutes. These simulations were associated with large mitochondria rescalings concomitant with large ER rescalings, demonstrating a net emptying of the ER stores into the mitochondria.

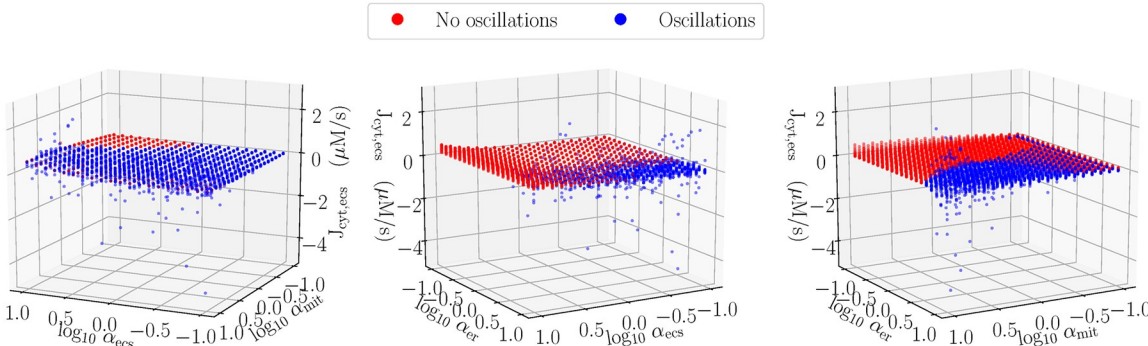

**Fig 7. Period-averaged flux from mitochondria to cytosol with presence of oscillations indicated.** Approximately 8% of simulations had net flux into the mitochondria at the end of 30 minutes. The majority had flux out of the mitochondria, with oscillatory simulations exhibiting larger average fluxes.

In the majority of simulations, there is a net emptying of mitochondria. This emptying proceeded at a slow rate in the absence of oscillations. In the presence of oscillations, larger net fluxes can be observed, with the caveat that the oscillations can result in periods where the fluxes reverse, as present in the simulation presented in Fig 3. Relative to the mitochondrial matrix, outward fluxes imply calcium dissolution, proton consumption, and ATP production. Inward fluxes result in the opposite.

## Discussion

We have presented a model of calcium dynamics within smooth muscle vascular cells incorporating relevant endoplasmic reticulum, mitochondrial, and plasma-membrane dynamics that participate in responding to elevations in calcium concentration. Using this model, we demonstrated macroscopic vascular behavior consistent with the hour-long disruption in neurovascular coupling following cortical spreading depolarization.

On the macroscopic level, the phenotype we describe is the post-acute dynamics seen in the pial artery diameter measurement of Figure 4A of [12]. Following acute CSD, over the period of a few minutes, the diameter drops from a prior resting state to a persistent constrictive state lasting for over half an hour. While the vessel is constricted, small-amplitude oscillations are visible, consistent with the result of our model in Fig 4. Note that the actual vascular radius is a function of the fluid pressures within the vasculature, the smooth muscle contractile force, and the tissue pressure. In our simulations, we only consider the contractile force.

Our chief modification to prior models has been the explicit handling of calcium buffering. Many prior studies assumed that a fixed proportion of calcium is buffered [51, 60, 71]. This assumption is an approximate form of the rapid buffer approximation that we use in this study. In particular, we distinguished between free and buffered calcium in the endoplasmic reticulum (ER). In doing so, we account for the finite capacity of the ER. The driving hypothesis of our paper is that solubility is essential for determining the recovery of calcium dynamics as affected by mitochondria after CSD. Solubility limits the rate of extrusion by reducing the active free ionic calcium concentration.

### Assumptions, limitations, and alternative mechanisms

We account for solubility in the mitochondria by invoking a quasi-steady state assumption on the concentration of free calcium under the rapid equilibration of dissolution and aggregation of calcium phosphate species. The actual chemical pathways underlying these transformations are complex but we assume that they are rapid relative to the rate of exchange mechanisms present in the mitochondrial membrane.

In formulating our model, we inherited some assumptions of prior work [14, 51, 55, 72, 73, 74]. We relied on prior models for understanding the dynamical implications of elevated mitochondrial calcium and on measurements on mesenteric smooth rather than cerebral smooth muscle cells. In some cases, we have tuned the parameters to reflect this change, for instance, by tuning down the activity of RyR relative to IP$_3$R. A major challenge of quantitative study in this field is in parametric uncertainty, as many reported values in the literature are based on the fitting of under-determined systems. We have attempted to put ourselves in a physiologically reasonable parameter regime from which we did exploration. However, this work would benefit from quality cell-specific measurements of the enclosed mechanisms.

Prior models distilled the biological variability in vascular smooth muscle calcium dynamics into a few most-salient mechanisms which we replicate in our model. We made the simplifying assumptions of ignoring voltage-dependent calcium re-entry and the dependence on concentrations of other ions. Hence, our estimate of the recovery timescale may be low due to

the fact that additional calcium entry or depletion of cofactors would increase the recovery time. Additionally, we neither modeled the energetic cascade behind ATP formation nor accounted for ATP depletion. This choice should also lead to an underestimate of recovery time. We believe our model represents a reasonable lower bound for vascular mitochondrial calcium recovery from CSD.

In our manuscript, we present a possible calcium-centric mitochondrial mechanism for vascular dysfunction. It should be noted, however, that there are other potential mechanisms of vascular origin that would cause long-lasting vasoconstriction. Since actin-myosin unbinding requires ATP, it could be the case that the smooth muscle cells are ATP starved. This starvation could be caused also through mitochondrial dysfunction resulting from mPTP opening.

Finally, in the current work, we mainly focus on proposing a simple model for the hour-long oscillation in the one cell after the actue phase CSD. In order to consider the wave propagation along the tissue, the spatial variation needs to be taken into consideration. A partial differential equations model like in [75] could help to connect the macro scale wave propagation with micro cytosol calcium oscillation.

## Physiological implications

Using our model, we investigated the expulsion of excess calcium from the mitochondrial matrix conditional on the prior formation of calcium phosphate clusters. Prior experimental evidence [40] has shown that the clustering process can be rapid within mitochondria, with the saturation concentration being near the concentrations of $[Ca^{2+}]_{mit}$ where we observe oscillations in our model.

Over half an hour, the simulation in Fig 4 is able to clear approximately 80 $\mu$M of calcium from the cell into the extracellular space. While calcium oscillations are physiologically important, our model shows they are unimportant for determining the timescale of calcium elevation within the mitochondria. Rescaling the mitochondrial and plasma membrane fluxes, as shown in Fig 6, modulates the recovery timescale whereas rescaling the ER flux has minimal effect, regardless of oscillations.

CSD is an universal phenomenon associated with many deleterious disorders. Its effects are both immediate and long-lasting. Much attention is paid to the role of neurons and glial cells in CSD. This manuscript is intended to shine a light on the vasculature itself as an active participant in its own dysregulation. Understanding the mechanisms behind the subacute CSD phase can aid in the development of treatments.

## Possible protective effect

Calcium phosphate dissolution in the matrix directly consumes free protons, where approximately three protons removed from the matrix equates to a single ATP molecule. Hence, the rate of ATP production resulting from calcium phosphate dynamics is proportional to the flux shown in Fig 7. When calcium flux out of the matrix is positive, the proton gradient is reinforced, supporting the production of ATP. Conversely, when the flux is negative, the proton gradient is compromised. If too many protons accumulate in the matrix, the $F_0F_1$-ATP synthase reverses and the enzyme consumes ATP rather than produces it.

On average, the flux out of the matrix is positive. Hence, not withstanding other negative effects on calcium dynamics, the dissolution process is partially protective in generating ATP through oxidative machinery, even in the absence of oxygen. However, as seen in Fig 3, due to oscillations, there can be periods where the matrix calcium triggers calcium re-entry, working against the proton gradient.

## Acknowledgments

Authors would like to thank Professors R.S. Eisenberg (Rush Medical University) and Dr. J.M. Han (NIDDK) for beneficial discussions.

## Author Contributions

**Conceptualization:** Joshua C. Chang, KC Brennan, Huaxiong Huang.

**Formal analysis:** Shixin Xu, Joshua C. Chang, Carson C. Chow.

**Funding acquisition:** Carson C. Chow, Huaxiong Huang.

**Investigation:** Shixin Xu, Joshua C. Chang, Carson C. Chow, Huaxiong Huang.

**Methodology:** Shixin Xu, Joshua C. Chang.

**Project administration:** Joshua C. Chang, Carson C. Chow, Huaxiong Huang.

**Software:** Shixin Xu, Joshua C. Chang.

**Supervision:** Carson C. Chow, Huaxiong Huang.

**Validation:** Shixin Xu, Joshua C. Chang, Carson C. Chow, Huaxiong Huang.

**Visualization:** Shixin Xu.

**Writing – original draft:** Shixin Xu, Joshua C. Chang, Huaxiong Huang.

**Writing – review & editing:** Shixin Xu, Joshua C. Chang, Carson C. Chow, KC Brennan, Huaxiong Huang.

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
