## [Decision Letter · Decision Letter 0]

4 Nov 2019

Dear Dr Chang,

Thank you very much for submitting your manuscript 'A mathematical model of mitochondrial calcium-phosphate dissolution as a mechanism for persistent post-CSD vasoconstriction' for review by PLOS Computational Biology. Your manuscript has been fully evaluated by the PLOS Computational Biology editorial team and in this case also by independent peer reviewers. The reviewers appreciated the attention to an important problem, but raised some substantial concerns about the manuscript as it currently stands. While your manuscript cannot be accepted in its present form, we are willing to consider a revised version in which the issues raised by the reviewers have been adequately addressed. We cannot, of course, promise publication at that time.

Sincerely,

Tim David

Guest Editor

PLOS Computational Biology

Feilim Mac Gabhann

Editor-in-Chief

PLOS Computational Biology

[LINK]

Reviewer's Responses to Questions

**Comments to the Authors:**

Reviewer #1: The review is uploaded as an attachment with the title "PLOS Comp. Bio. Review - Xu et al.docx"

Reviewer #2: The review is uploaded as an attachment.

Reviewer #3: Review of “A mathematical model of mitochondrial calcium-phosphate dissolution as a mechanism for persistent post-CSD vasoconstriction”

By Xu S, Chang JC, Chow CC and Huang H.

Summary: This manuscript assembles a computational model to explain elevated cytosolic calcium levels and clearance in vascular smooth muscle cells (VSMCs) of the cerebral vasculature after a cortical spreading depolarization (CSD) event. This model describes calcium transport through the cellular membrane, between the cytosol and the endoplasmic reticulum, and between the cytosol and the mitochondrial matrix. This model is able to describe the hour long timescale for cytosolic calcium to return to normal after a CSD event.

Overall Comments: The authors have built a simplified model to represent the calcium levels in VSMCs after CSD and have clearly described their model in the text. They have also effectively varied parameters to try to understand the important parts of the model. However, no experimental data is shown and in none of the figures the actual rise and decay of calcium over the post CSD time period is shown. Parameters are clearly presented along with their sources which enables reproducibility however searching for the code under the username joshchang on github did not yield the code for inspection.

Specific Comments:

1) No data is represented to challenge the results of the model.

2) Make sure that code is available on github at least and it is suggested to take advantage of the code check that is an option with PLoS Comput Biol.

3) It is stated that in figure 2 the net movement of the calcium is from the mitochondria to the cytosol to the extracellular space however the net flux from mitochondria to cytosol in figure 2 is essentially zero at the resolution of the figure.

4) The simulations of figures 3 and 4 show some variability with no apparent pattern. Why is this the case if you simply have a set of ODEs modeled here?

5) The authors have made clear that local regulatory mechanisms have been omitted in this model. However in the discussion the interaction between these Ca2+ transients and those generated by the myogenic response or shear mediated dilation should be discussed.

Specific Minor comments

1) Figure 4 shows the clearance of Ca2+ in panel 1 and the release of Ca2+ from the mitochondria in panel 2 but does not resolve the time course of Ca2+ in the cytosol over the 30 minute timescale depicted.

2) Figure 5 is not really clear. Suggest one 3D view with 3 views comparing each pair of parameters (top, front and side).

3) On page 15 the authors suggest that cell-specific measurements would be beneficial for the identification of this model. Can the authors suggest what measures would have the most utility to advance this work?

4) Mitochondrial permeability transition pore opening leads to mitochondrial dysfunction and cell death. It is assumed in this model that the VSMCs recover. Is this tractable?

Pdf version attached

Reviewer #4: The review is uploaded as an attachment

Reviewer #5: see attachment

**Have all data underlying the figures and results presented in the manuscript been provided?**

Reviewer #1: Yes

Reviewer #2: Yes

Reviewer #3: Yes

Reviewer #4: Yes

Reviewer #5: Yes

PLOS authors have the option to publish the peer review history of their article (what does this mean?). If published, this will include your full peer review and any attached files.

Reviewer #1: No

Reviewer #2: No

Reviewer #3: Yes: Brian E. Carlson

Reviewer #4: No

Reviewer #5: No

---

## [Decision Letter · Decision Letter 1]

28 May 2020

Dear Dr. Chang,

We are pleased to inform you that your manuscript 'A mathematical model for persistent post-CSD vasoconstriction' has been provisionally accepted for publication in PLOS Computational Biology.

Best regards,

Tim David

Guest Editor

PLOS Computational Biology

Feilim Mac Gabhann

Editor-in-Chief

PLOS Computational Biology

Reviewer's Responses to Questions

**Comments to the Authors:**

Reviewer #1: Reviewer 1 comments for the resubmission is attached

Reviewer #2: I am satisfied that the authors have addressed my concerns.

Reviewer #5: I found all of my topics of criticism ansered satisfyingly.

**Have all data underlying the figures and results presented in the manuscript been provided?**

Reviewer #1: Yes

Reviewer #2: Yes

Reviewer #5: None

PLOS authors have the option to publish the peer review history of their article (what does this mean?). If published, this will include your full peer review and any attached files.

Reviewer #1: No

Reviewer #2: No

Reviewer #5: No

---

## [Editor Report · Acceptance letter]

30 Jun 2020

PCOMPBIOL-D-19-01345R1 

A mathematical model for persistent post-CSD vasoconstriction

Dear Dr Chang,

I am pleased to inform you that your manuscript has been formally accepted for publication in PLOS Computational Biology. Your manuscript is now with our production department and you will be notified of the publication date in due course.

With kind regards,

Laura Mallard
